# Experiences and Perceptions of Trans and Gender Non-Binary People Regarding Their Psychosocial Support Needs: A Systematic Review of the Qualitative Research Evidence

**DOI:** 10.3390/ijerph18073403

**Published:** 2021-03-25

**Authors:** Edward McCann, Gráinne Donohue, Michael Brown

**Affiliations:** 1School of Nursing and Midwifery, Trinity College Dublin, Dublin, Ireland; donohuga@tcd.ie; 2School of Nursing and Midwifery, Queen’s University, Belfast BT9 7BL, UK; m.j.brown@qub.ac.uk

**Keywords:** transgender, mental health, psychosocial, supports, experiences, qualitative evidence

## Abstract

People who identify as trans and gender non-binary experience many challenges in their lives and more interest is being paid to their overall health and wellbeing. However, little is known about their experiences and perceptions regarding their distinct psychosocial needs. The aim of this systematic review is to critically evaluate and synthesize the existing research evidence relating to the unique psychological and social experiences of trans people and identify aspects that may help or hinder access to appropriate psychosocial interventions and supports. The PRISMA procedure was utilized. A search of relevant databases from January 2010 to January 2021 was undertaken. Studies were identified that involved trans people, and addressed issues related to their psychosocial needs. The search yielded 954 papers in total. Following the application of rigorous inclusion and exclusion criteria a total of 18 papers were considered suitable for the systematic review. Quality was assessed using the MMAT instrument. Following analysis, four themes were identified: (i) stigma, discrimination and marginalization (ii) trans affirmative experiences (iii) formal and informal supports, and (iv) healthcare access. The policy, education and practice development implications are highlighted and discussed. Future research opportunities have been identified that will add significantly to the body of evidence that may further the development of appropriate health interventions and supports to this population.

## 1. Introduction 

People whose sex assigned at birth differs from their gender identity are known as transgender or gender diverse. Transwomen were assigned male at birth and transmen female. Non-binary people, including non-conforming or genderqueer, do not identify with a specific gender [1]. The authors use trans throughout this paper as an umbrella term to describe people whose gender is, dissimilar to or does not sit comfortably with, the sex assigned to them at birth [2].

An estimated 1.4 million people in the US identify as trans [3]. In the United Kingdom (UK) the most accurate estimate is 1 per cent of the population or 600,000 people [4]. A significant number of trans and non-binary people continue to experience many challenges in their daily lives. The cumulative effects of minority stress can have a significant and detrimental effect on the trans and non-binary person’s psychosocial health and well-being and negatively impact upon the individual’s quality of life [5,6]. Mental health issues are more prevalent in this population with 1:2 trans people being diagnosed with depression and anxiety compared to 1:5 in the general population. Additionally, suicidal thoughts and attempts are significantly higher than in the general population with as many as 41% of trans people attempting suicide [7]. 

Higher mental health morbidity rates have been attributed to discrimination and victimization experiences as well as family rejection and estrangement issues [8]. Experiences of violence including intimate partner violence was also associated with higher rates of mental health diagnoses [9]. Heavy alcohol or Illicit drug use, having a disability or being homeless were also contributory factors [10,11]. Access to gender affirming care was associated with lower prevalence rates of suicidal thoughts and attempts [12]. 

The COVID-19 pandemic has exacerbated ongoing mental health issues and concerns for trans and non-binary people. Disrupted or reduced support to people who are trans and non-binary has been associated with increased psychological distress during the pandemic [13]. In one study, 55% of respondents reported reduced access to gender-affirming care. More than half screened positive for depression and 45% for anxiety. Suicidal ideation was also experienced by a significant number of study respondents [14]. 

Mental health services, supports and interventions tailored and responsive to the specific needs of the trans and non-binary population is variable depending on the context and jurisdiction [15,16]. Access to services can be problematic with trans individuals reporting a reluctance to use services for fear of discriminatory practices and further marginalization and social exclusion [17]. Necessary psychological supports and interventions, including access to a range of talking therapies, particularly with a focus on minority stress should be widely available [18]. However, such interventions need to be delivered by knowledgeable and culturally competent practitioners skilled in trans-affirmative approaches [19]. There still remains a distinct lack of studies that systematically evaluate these interventions [20]. In recognition of the greater risk to the well-being of trans and non-binary young people compared to that of their non-trans peers, digital health interventions have been developed to address potential health difficulties including psychosocial issues and concerns. These are seen as adjuncts to face-to-face services and include the use of web-based (e-health) of mobile (m-health) mental health interventions. Online activities and supports may address stigma, isolation, mental health issues and social concerns such as living in remote or rural communities thus promoting social inclusion [21,22]. Despite the disproportionately high mental health risks associated with minority stressors, trans and non-binary people can and do develop resilient traits that buffer against the negative effects of adverse experiences. The Transgender Resilience Intervention Model (TRIM) has been proposed that addresses, for example, social support, community belonging and family acceptance, as well as self-worth, pride and hope [23].

Evidently, there are significant challenges to the lives of trans and non-binary people. However, the psychosocial needs of trans and non-binary people remain largely unexplored. Therefore, the aim of this review was to systematically evaluate the best available evidence on the experiences and perceptions of trans and non-binary people regarding their psychosocial needs, and to identify aspects that may influence access to necessary psychosocial interventions and supports.

## 2. Methods

The objectives of the systematic review were to:(i)identify the experiences and perceptions of trans and non-binary people regarding their psychosocial needs,(ii)establish the psychosocial interventions and supports that are available to people who identify as trans and non-binary?(iii)highlight the best practice examples that exist.

### 2.1. Ethics Statement

The study is a systematic review of published research evidence therefore ethical approval was not required.

### 2.2. Search Strategy

A subject Librarian was enlisted to assist with the literature search strategy. The databases used in the search were CINHAL (EBSCO host), MEDLINE (Ovid) and PsycINFO (Ovid). The search strategy used in all of the electronic databases is shown in Table 1. Databases were searched up until 31 January 2021. Considered papers were written in English, peer reviewed and published after 2010. The research team specifically selected a ten-year time frame as it reflects the changes in legislation and policy over the past decade that has sought to respond to and recognize the human rights and equality agenda in relation to LGBQ and trans people. A hand search was conducted of reference lists and Google Scholar used to identify any other relevant papers. 

### 2.3. Inclusion and Exclusion Criteria

Included studies had used a qualitative approach and addressed the distinct experiences and perceptions of trans and gender non-binary people. Studies that identified psychosocial interventions and supports were considered for inclusion. Studies that did not specifically focus on trans or non-binary people or deal with the objectives of the review were excluded. 

### 2.4. Data Extraction

Following the removal of duplicate papers, two reviewers (EM and GD) screened the title and abstracts using the study inclusion criteria. The *Covidence* systematic review software (Veritas Health Innovation, Melbourne, Australia) was used in the process [24]. The reviewers retrieved and independently screened the full text papers. The reviewers then critically appraised the papers, and any disagreements were resolved following discussion. 

### 2.5. Data Synthesis

The synthesis of the research literature was carried out as part of the systematic review process. Firstly, the data was firstly independently thematically analyzed by each member of the research team to identify the preliminary emergent themes across the included studies. Next, the research team collectively appraised the included studies and undertook a detailed and comprehensive coding and analysis. Then, the research team collectively identified the key emerging concepts that were then grouped into possible themes. The themes across the included studies were then compared to identify contrasting similarities and differences and following this rigorous process the final themes were discussed, verified and agreed by the research team (EM, GD and MB) [25].

### 2.6. Quality Assessment

Two reviewers carried out the appraisal of the included studies (Table 2). The qualitative component of the Mixed-Methods Appraisal Tool (MMAT) was used in the assessment of the quality of the included studies [26]. To determine the quality of the studies, a range of questions were applied. Each study could score ‘high’ ‘medium’ or ‘low’ depending on specific criteria. A total of sixteen studies achieved ‘high’ scores [27,28,29,30,31,32,33,34,35,36,37,38,39,40,41,42]. Medium scores were achieved by two studies [43,44]. All of the studies met the objectives of the review and were deemed suitable for inclusion in the systematic review. 

Critical appraisal questions were as follows:Is the qualitative approach appropriate to answer the research question?Are the qualitative data collection methods adequate to address the research question?Are the findings adequately derived from the data?Is the interpretation of results sufficiently substantiated by data?Is there coherence between qualitative data sources, collection, analysis and interpretation?

## 3. Results

### 3.1. Search Results

The searches revealed 954 hits across all the databases. Search limiters of peer reviewed empirical studies and written in English were applied and duplicates removed leaving 687 papers for abstract screening. Following this step, a total of 52 full text papers were assessed for eligibility using the specified inclusion criteria. A hand search was also conducted of the reference lists of the identified papers leaving a total of 18 papers for the review. The Preferred Reporting Items for Systematic Reviews and Meta-analyses Statement (PRISMA-S) process was followed, and a flow chart is provided (Figure 1) that contains the results of the searches [45]. 

### 3.2. Study Characteristics

The 18 studies that addressed the aim and objectives of the review are contained in Table 3. The majority of studies were undertaken in the USA (*n* = 10). The remaining studies were from Canada (*n* = 3), Ireland (*n* = 2), Australia (*n* = 1), China (*n* = 1), Columbia (*n* = 1), India (*n* = 1), Mexico (*n* = 1) and Sweden (*n* = 1). Study participant numbers ranged from 4 to 45 and the recruitment strategies appeared robust. The ages were 15 to 60 years. All of the studies used qualitative research methods that explored and documented the experiences and perceptions of trans and gender non-binary people in relation to their distinct psychosocial support needs.

### 3.3. Thematic Analysis

Following the systematic analysis of the studies, four themes were identified: 

(i) Stigma, discrimination and marginalisation (ii) transgender affirmative experiences (iii) formal and informal supports, and (iv) healthcare access.

#### 3.3.1. Stigma, Discrimination and Marginalisation 

People who are trans or gender non-binary can experience many challenges in their daily lives that can significantly impact upon their general and psychosocial well-being. Healthcare experiences vary worldwide where transgender-specific needs and requirements are being increasingly recognised and specific stigma-related barriers to healthcare identified [27,29,32,40,42,43]. The impact of discrimination and prejudicial societal attitudes and behaviours can result in issues related to *minority stress* further compounded by marginalisation and social exclusion experiences [30,31,33,34,38,44]. Furthermore, psychosocial challenges surrounding loss of autonomy, increased social isolation and loneliness as a result of the COVID-19 pandemic is now gaining attention, notably for older transgender adults [27]. Related to societal exclusion is the phenomenon of erasure where transgender identity or existence is ignored, minimised or denied. In some of the studies, this had a significant impact upon the experiences of transgender people, their own perceptions and the attitudes of others towards them and their potential visibility, inclusivity and human rights [31,37,38,39,41]. A significant number of transgender people have been subjected to physical violence and other forms of abuse and some have been disowned by their families. These experiences can have a profound and detrimental effect on the well-being of the individual, and their quality of life and mental health outcomes [30,36,42]. There are well documented and significant experiences of depression, anxiety and suicidality in this population [27,33]. These issues and concerns are more pronounced and problematic for transgender people held within the prison system. Many experienced restricted access to healthcare due to stigma and discrimination where existing policies reinforce gender binary, support sex-segregated prisons, and employ prejudiced, biased and inexperienced prison staff. Such conditions limited the access to appropriate supports and healthcare and had a negative impact on transgender participants’ overall health and psychosocial circumstances whilst imprisoned [30]. 

As well as access to sufficient and appropriate healthcare and supports, the apparent ‘mistreatment’ of this population can have significant implications in terms of human rights and rights-based care and supports [34,43]. For instance, in one study, undertaken in Columbia, where access to appropriate supports was denied, some participants had injected toxic substances into their bodies in an attempt to achieve specific bodily alterations, often with detrimental outcomes. Others utilised contraband self-administered hormone treatments without medical supervision or guidance. A significant number of participants (*n* = 20) in another study experienced ill-treatment and multiple human rights violations including ‘torture’ and ‘medical negligence’. All participants (*n* = 28) had been denied access to mental and medical healthcare and had experienced gender-based violence. Most had become displaced and estranged from their families. All had experienced depression, anxiety and had been involved in drug use [36]. 

Although the challenges associated with transgender experiences were significant, some study participants provided positive accounts of their experiences and circumstances. They had developed resilient features such as implementing coping strategies that assisted in character building, developing a stronger sense of identity and the growth of self-esteem through experiencing and dealing with adverse situations [28,37,41].

#### 3.3.2. Transgender Affirmative Experiences

An important empowering and affirming stance identified in the review was the non-pathologizing of gender variance and not viewing transgender experiences as a mental illness [35]. Three of four participants in one study had been taken to specialists and did not receive a diagnosis to expedite their access to transition-related care. Instead, they experienced criticism, judgement and rejection. Many were of the view that it would have been helpful to have ‘received a diagnosis’ and their family given the necessary information and supports to enable them to better understand the processes involved such as medical interventions and the use of hormones [43]. Positive parental supports and behaviors resulted in better mental health and well-being for some transgender young people due to the perceived psychosocial benefits. Supportive examples included identity affirmation, self-education, emotional support and instrumental support such as buying gender affirming clothes, make-up or binders [31].

Younger adults in Canada found it difficult to access specialized trans-affirmative care, especially when located in major urban centers. Youth that could access the necessary mental health and medical supports in these centers reported that they felt valued and affirmed. However, study participants also highlighted the scarcity of trans-friendly professionals and a general lack of awareness and training of practitioners within mainstream health service [37]. In several studies, the importance of finding knowledgeable, confident and culturally competent practitioners and support services able to contribute to the person’s physical health and psychosocial well-being was identified. Practitioner attributes including warmth, openness, rapport, respect, trust and acting professionally were all identified as important necessary. It was also identified that practitioners required to develop sensitivity and cultural awareness to respond fully and holistically to the support needs of transgender people and their families [38,40,41,43].

#### 3.3.3. Formal and Informal Supports

Members of the transgender community have distinct support needs that require health professionals to possess the appropriate knowledge, skills, attitudes and competencies to enable them to respond effectively. It has now been established that a welcoming healthcare environment with supportive and respectful healthcare professionals can contribute to the overall well-being and health of transgender people [37]. Two papers within this review explored experiences of psychosocial interventions as a way to identify the specific therapeutic challenges that transgender people can encounter [28,35]. In Benson [28], problems in the practice of psychotherapy were related to the belief that therapists were not generally well-informed about the distinct needs of transgender people. Participants discussed the need for transgender-informed therapists, as their experiences evidenced that many therapists were not adequately educated, often relying on clients to provide information and education about the needs of transgender people. Transgender people felt supported when mental health professionals were non-judgmental, affirmed their gender and appeared to care about their unique experiences. Specifically, moments of acceptance were recalled by participants when mental health providers validated their gender identity [28]. Again, in Mizock & Lundquist’s study, the result of psychotherapy ‘missteps’ meant that transgender or gender non-conforming clients (TGNC) were left with the burden of educating their therapists on gender identity, necessary to receive effective treatment. Added to this was the unhelpful experience of the psychotherapist adopting the position of gatekeeper; this was experienced by TGNC people as overly focused on controlling access to gender-affirming medical resources [35]. Supporting these findings, one study found that non-affirmative experiences in accessing mental health services, contributed to poor clinician–patient relationships with TG populations, impacting upon attrition [29].

In terms of informal supports, a study exploring parental support amongst transgender youth (*n* = 24), found that rejecting and mixed response parental behaviors contributed to a range of psychosocial problems, for example, depression and suicidal ideation, while supportive behaviors increased positive wellbeing [31]. Supportive behaviors included instances where parents made independent efforts to learn about transgender issues and help their child obtain gender-affirming health care. Rejecting behaviors included instances when parents refused to use their child’s chosen name or pronouns, thereby failing to show empathy when their child struggled with gender-identity-related challenges [31].

Alongside familial or parental involvement, peers are also recognized as an important source of support for transgender individuals [32,37,39]. Strauss et al. [39] explored the use of games as a gateway to bolster well-being in transgender youth. Apps were reportedly identified as valuable due to the mental health management skills that they teach the individual, such as coping mechanisms and the promotion of self-care. Participants voiced that a positive feature of many games is the ability to express their affirmed gender vis-à-vis the games played alongside peers [39]. In another study, also investigating youth transgender experiences, researchers identified that supportive psychological services can enhance a transgender youth’s capacity to face adversity and build resilience [37]. Additionally, and beyond these institutional spaces, transgender youth also voiced that their relationships with their families and with other close social or peer groups significantly influenced their overall well-being [37]. Finally, the study by Knutson et al. illustrated the ways that peer support can assist transgender people in rural areas by encouraging each other to embrace their gender identities, seeing their own value, and using their self-awareness as a source of motivation to seek care, despite the inherent challenges [32].

#### 3.3.4. Healthcare Access

Transgender people experience significant health disparities and often require healthcare interventions as part of their ongoing care and support [29,30,32,33]. It is important therefore that healthcare experiences are positive, trans-affirmative and informed by past transgender experiences. In a study investigating the healthcare experiences of incarcerated transgender women participants described an institutional culture in which their feminine identity was not recognized, and institutional policies acted as a form of structural stigma that created and reinforced gender binary [30]. Some participants saw these barriers as a result of bias, others attributed barriers to the often-limited knowledge or inexperience caring for transgender patients on the part of service providers. Lack of knowledge on the part of healthcare providers was also experienced as a barrier in Delaney and McCann’s study, whereby alongside misinformation, the pathologizing of the transgender identity led to non-affirmative healthcare experiences, affecting attrition [29].

Lack of recognition of gender identity can result in barriers to care, including clinics specializing in gender-affirming healthcare [33]. In this study, gender-queer and non-binary youth felt misunderstood by providers who approached them from a binary transgender perspective and consequently did not receive care that was sensitive to their non-binary identities. This resulted in a forced conformity to binary medical narratives throughout healthcare interactions [33]. For Knutson et al. transgender people in rural settings described accessing healthcare as akin to ‘putting a target on your back’ (pg.114) [32]. Participants experienced the burden of quality control for transgender healthcare services which often fell to the person accessing the service. Despite the recognition that they may encounter discrimination, prejudice and mistreatment while accessing care, participants also expressed commitment to pursuing appropriate care. This is an important finding in the context of the evidence that some healthcare settings can be oppressive spaces for transgender people [32]. 

## 4. Discussion

While there is an international focus on social justice and equality for all, it is evident that this is not the reality for many trans and non-binary people who continue to experience stigma, discrimination and marginalization in their daily lives. Others also experience violence and assaults that significantly affects their safety and impacts on their health and well-being. Where trans people and non-binary have positive and affirming experiences, from regarding their identity by parents, families and friends, with access to formal and informal networks, a positive impact on their health and well-being, including mental health, has been identified [29,37,41]. Conversely, rejecting behaviors that fail to provide positive affirmation, have a negative effect [30,31]. Therefore, building resilience and self-coping and adaptation skills have also been shown to benefit mental health and well-being. However, from a health perspective, trans and non-binary people report barriers when seeking access to appropriate healthcare, a situation that further contributes to their health inequalities and poor health outcomes [43,44]. The situation is compounded as many health professionals have limited knowledge and understanding of the health and specific support needs that require to be addressed. It is therefore apparent from the findings arising from this review that there remain significant implications for healthcare. Further actions are necessary from a policy, education and practice development perspective and opportunities for further research to ensure that the psychosocial needs of trans and non-binary people are effectively identified and met. 

### 4.1. Policy 

Due to the distinct needs of trans and non-binary people in relation to the wider gay communities, there is requirement to ensure that policies are reflective of their concerns, thereby reducing the health inequalities gap. Policies with a focus on the needs of LGBTQI+ people, for example, need to ensure that the stigma and discrimination that continues to be a reality in the lives of many trans and non-binary people are acknowledged, and strategies developed and implemented [46,47,48]. Recent decades have seen increased attention on the mental health requirements of the general population and the investment in psychological therapies and supports [49]. As evidenced in this review, policies need to promote cultural competence and confidence by ensuring that practitioners across health services have the knowledge and skills necessary to recognize and respond sensitively to the needs and concerns of trans and non-binary people [28,29,34,36,43]. Particular policy responses are required by mental health services, given the scope and extent of the psychological needs of many trans and non-binary people, thereby aiming to ensure that assessment, treatment, interventions and supports are responsive to their needs and concerns [27,37,40,42]. Primary care services have an important role to play with an opportunity to ensure that policies are reflective of and take account of the needs of trans people and non-binary due to the extent of their health needs [50,51]. Likewise, sexual health policies also have a role to play in responding to the sexual health needs of this population, recognizing that some experience abuse, harm and exploitation and some people may engage in prostitution and survival sex [52,53]. Additional challenges in light of the emerging evidence of the long-term effect of the COViD-19 pandemic needs to be reflected in future policy developments, inclusive of the needs of LGBTQI+ people [27]. 

### 4.2. Education and Practice Development

In response to the evidence of the often-limited knowledge and confidence when providing care and support for trans people and non-binary, there is a pressing need for practitioners to access education and practice development opportunities. Evidence-informed education is required as it is apparent that the healthcare experiences of many trans and non-binary people can be unsatisfactory. Some trans and non-binary patients report having to educate health professional about trans issues often before their needs are assessed and addressed [40]. While opportunistic education provided by trans and non-binary patients may in some circumstances be appropriate, the primary focus must be on culturally sensitive and responsive care provided by the health professional with the prerequisite knowledge and skills [28,34,37,41]. There are opportunities for education collaborations involving trans and non-binary people to enable health professionals to learn from their knowledge and experiences, thereby aiming to ensure person-centered healthcare experiences [54]. To grow and develop the knowledge and confidence of the future health workforce, there is also an opportunity to incorporate education about the needs of LGBTQI+ people, including trans and non-binary issues, within undergraduate programs for doctors, nurses, midwives, social workers and other health and social care students [55]. Additionally, from a postgraduate education perspective there are opportunities to develop shared learning opportunities across professional disciplines, such as health, teachers, social care, the police and criminal justice organizations and non-government organizations, for example LGBTQI+ and trans and non-binary community support groups. As education initiatives respond to the COVID-19 pandemic and the long-term effects, there is a need to ensure the care and support needs of LGBTQI+ and trans and non-binary people are reflected in relevant curricula. Shared learning opportunities could involve trans people in the co-design, delivery and evaluation of education initiatives, thereby aiming to identifying the wider impact on their lives and health and well-being [56].

### 4.3. Practice 

The psychosocial needs of trans and non-binary people are largely unexplored and therefore the evidence-base available to inform practice is an area requiring further development to ensure that care and support of person-centered and outcome based. What is evident from the current review is that the experiences of many trans and non-binary people when accessing healthcare and specifically mental health services is unsatisfactory [32,33,34,36]. The situation is further compounded by the prevailing heteronormative assumptions that continue to pervade within health services [29,42,44]. For example, some trans people have reported sexual orientation and gender identity change efforts, sometimes known as conversion or ‘reparative therapy’, whereby health professionals seek to prepress and alter a person’s sexual orientation or gender identity [57]. Therefore, practitioners must recognize their position regarding possible discriminatory behaviors and negative attitudes towards trans and non-binary patients and proactively eliminate them, thereby demonstrating professionalism and regulatory accountability [58]. The practice of health professionals needs to be firmly set in the context of anti-discriminatory practice and clearly aligned to social justice and equality of access and health outcomes, irrespective of gender identity [59]. From a mental health practice perspective, there is a need to ensure that the assessments, interventions and supports provided to trans patients is culturally competent and responsive to their concerns [60]. Physical and mental ill-health is common in many trans and non-binary people, and there is a need for practitioners to respond to the emerging evidence of the long-term impact and effects of COVID-19 health needs of on this population [61]. Additionally, practitioners can also offer education, advice and support to parents, families and friends of trans and non-binary people, thereby assisting in their understanding and acceptance of often new and complex issues. Social isolation can be an issue for some trans and non-binary people and therefore there is scope for mental health professionals to work with community organizations and networks to ensure there is effective local services available to provide information and necessary supports [62]. From the perspective of evidence-based therapies, given the scope and extent of the mental health needs of trans and non-binary people, there is a need to ensure that professionals delivering psychological interventions, such as counselling and psychotherapies have the necessary attitudes, values and skills to address their specific concerns [63,64].

## 5. Strengths and Limitations 

There are several strengths to the current review. The positive role of parents, families and friends in relation to gender affirming supports is an important finding and is an area requiring further enquiry. Providing accessible community spaces with access to information, advice and support is another strength identified from the findings, thereby helping to promote resilience and self-coping abilities. The immediate and possible long-term effects of COVID-19 are starting to be realized and this review has included the first study to focus on the needs of older trans and non-binary people, an issue that needs to attract attention in the future. There are several limitations that need to be acknowledged. A majority of the studies were undertaken in North American and European countries, where the experiences of trans and non-binary people may not be reflective of those more widely. Therefore, some caution is required with the translation of the findings to different countries internationally, where legislation, policies and health service provision may differ. While the growing evidence is welcome, it is important to note that there was an absence of multi-centered national and international studies or longitudinal long term psychosocial needs of transgender people and subpopulations within the trans and non-binary communities such as people with disabilities and people of color. While seeking to be rigors throughout the review process, the authors acknowledge the possible subjectivity and biases that may exist. 

## 6. Future Research 

While this review presents the current evidence-base of the psychosocial needs and concerns of trans and non-binary people, there are opportunities to grow and develop further research in this area. While there is a need for further LGBTQ+ research, there is an opportunity to focus specifically on the needs of trans and. non-binary people as distinct from other populations that sit under the LGBTQ+ ‘umbrella’. From a research perspective this is necessary as the needs of these subpopulations are not homogeneous and therefore each warrant specific attention. Research focusing on the views and experiences of trans and non-binary people when accessing health services, including mental health services, is an area ripe for further enquiry, thereby growing the evidence-base of their needs and the responses required. To date, one research study has focused on the impact on COVID-19 on older trans people. There is therefore an important emerging area for future research, with LGBTQ+ and trans and non-binary people included within study samples and also studies focusing on their specific needs and concerns. The existing research evidence-base in this area are all single-center, utilizing cross-sectional designs and convenience samples, primarily due to the challenges of accessing trans and non-binary study participants. There is therefore an opportunity to develop national and international collaborations that allows for larger samples and enables comparisons and identification of views, experiences, needs and service developments, thereby growing the research evidence-base of what works for trans and non-binary people. 

## 7. Conclusions

It has become increasingly apparent through this systematic review of the research evidence that trans and non-binary people have distinct psychosocial needs. There is an evolving body of research evidence detailing the support experiences of this population and whilst there are some pockets of good practice, it is the responsibility of healthcare professionals to also learn from the non-affirmative experiences that people have documented in accessing supports. Discriminatory experiences in accessing healthcare for trans people and non-binary has compounded the *minority stress* already felt by this population. This has led to profound feelings of erasure, whereby trans and non-binary identity is either ignored, minimized or denied in healthcare practices. Trans and non-binary gender affirming experiences on the other hand, have the potential to restore autonomy and impact greatly upon attrition rates, thus improving healthcare outcomes.

The limitations of psychosocial supports lie predominantly in the absence of services that are embedded within a knowledgeable and culturally sensitive workforce. This review highlights that future support services should not rely on the trans and non-binary person to shoulder the burden of educating practitioners on their specific needs. Rather, resources should be directed into nurturing a culturally competent workplace, that can meet the specific and diverse needs of the trans community. It is recommended that to develop a service that meets the specific needs of the trans and non-binary populations, clear policies should exist that pro-actively communicate a non-discriminatory practice based on gender identity, expression and orientation. Failure to consider these important recommendations will force conformity to binary medical narratives and lead to a further alienation of trans individuals in accessing basic supports.

Therefore, to address these needs, it is necessary to provide and promote access to education and practice initiatives that build upon health professional’s confidence, skills and knowledge in working with trans and non-binary people. Future studies should build upon best practice for maintaining and increasing safe gender-affirmative healthcare.

## Figures and Tables

**Figure 1 ijerph-18-03403-f001:**
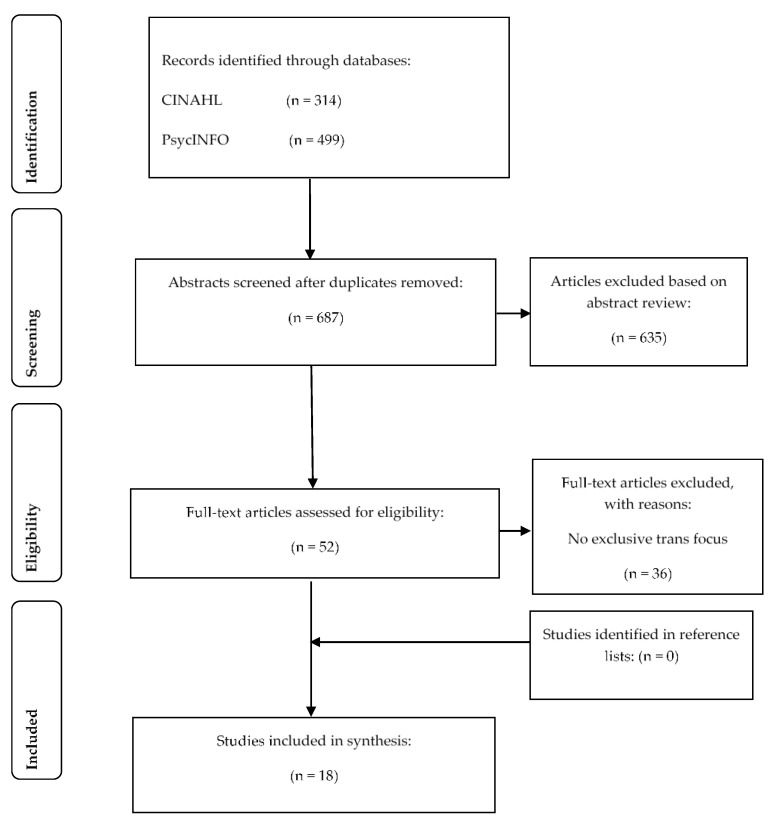
PRISMA Flow Diagram with search result.

**Table 1 ijerph-18-03403-t001:** Search results across all databases.

Search Code	Query	PsycINFO	MEDLINE	CINHAL
S1	transgender or transexual or gender dysphoria or gender non-conforming	11,514	10,711	6529
S2	mental health services or mental health care or psychosocial supports	178,523	87,825	144,409
S3	opinions or views or perceptions or experiences or qualitative	1,376,790	1,825,221	700,730
S4	S1 AND S2 AND S3	499	141	314
S5	Limiters: academic peer reviewed papers, written in English	392	141	308

**Table 2 ijerph-18-03403-t002:** Methodological quality of qualitative studies using MMAT (Hong et al. 2018).

Study	Q1	Q2	Q3	Q4	Q5	Quality Score
Banerjee & Rao (2021) [27]	Y	Y	Y	Y	Y	H
Benson (2013) [28]	Y	Y	Y	Y	Y	H
Delaney & McCann (2020) [29]	Y	Y	Y	Y	Y	H
Hughto et al. (2018) [30]	Y	Y	Y	Y	Y	H
Johnson et al. (2020) [31]	Y	Y	Y	Y	Y	H
Knutson et al. (2018) [32]	Y	Y	Y	Y	Y	H
Lykens et al. (2018) [33]	Y	Y	Y	CT	Y	H
McCann (2015) [34]	Y	Y	Y	Y	Y	H
Mizock & Lundquist (2016) [35]	Y	Y	Y	Y	Y	H
Real-Quintanar et al. (2020) [43]	Y	Y	CT	CT	CT	M
Ritterbusch et al. (2018) [36]	Y	Y	Y	Y	Y	H
Sansfaçon et al. (2018) [37]	Y	Y	Y	Y	Y	H
Santos et al. (2019) [38]	Y	Y	Y	Y	Y	H
Strauss et al. (2019) [39]	Y	Y	Y	Y	Y	H
Taylor (2013) [44]	Y	CT	Y	CT	CT	M
Vermeir et al. (2018) [40]	Y	Y	Y	Y	Y	H
von Vogelsang et al. (2016) [41]	Y	Y	Y	Y	Y	H
Yan et al. (2019) [42]	Y	Y	Y	Y	Y	H

Y = yes, indicates a clear statement appears in the paper which directly answers the question; N = no, indicates the question has been directly answered in the negative in the paper; CT = cannot tell, indicates there is no clear statement in the paper that answers the question.

**Table 3 ijerph-18-03403-t003:** Papers included in the review (*n* = 18).

Citation and Country	Aim	Sample	Methods	Main Results	Recommendations
Banerjee & Rao (2021) India [27]	To explore the lived experiences and psychosocial challenges of older transgender adults during the COVID-19 pandemic in India.	Transgender individuals (*n* = 10), aged > 60 years.	In depth individual interviews. Phenomenological analysis.	Categories identified were marginalization, the dual burden of “age” and “gender” and multi-faceted survival threats during the pandemic. Social rituals, spirituality, hope, and acceptance of “gender dissonance” emerged as the main coping factors, whereas their unmet needs were social inclusion, awareness related to COVID-19 and mental health care.	Older aged gender minorities are at increased emotional and social risk during the ongoing pandemic. The need for policy implementation and community awareness about their social welfare is vital to improving their health and well-being.
Benson (2013) USA [28]	To critically review historical views of transgender clients and to highlight experiences of transgender clients in therapy.	Transgender individuals (*n* = 7), aged 24–57 years.	In-depth individual interviews. Feminist phenomenology informed analysis.	Four themes emerged: the purposes transgender clients sought therapy for, problems in practice, therapist reputation, and transgender affirmative therapy.	Mental health professionals need transgender specific training to stand against a history of pathology and acquire the skills and sensitivity needed to best support transgender clients.
Delaney & McCann (2020) Ireland [29]	To explore the personal experiences of transgender people with Irish mental health services.	Transgender individuals (*n* = 4), aged 29–45 years.	Semi-structured individual interviews. Interpretative phenomenological analysis.	Three themes emerged: affirmative experiences, non-affirmative experiences and clinician relationship. Lack of information and non-affirmative experiences are contributing to poor clinician–patient relationships with transgender populations and impactingattrition.	Modules including information on the transgender community should be included in nursing curricula and supported by nursing management. These modules should support a gender-affirmative approach to the care of transgender populations. Future research should explore the feasibility of including transgender-specific education into the curriculum for training nurses in an Irish context as well as supporting nurse managers in this area.
Hughto et al. (2018) USA [30]	To assess the experiences of incarcerated transgendered women receiving physical, mental and transition-related healthcare in correctional settings and to document potential barriers to healthcare.	Transgender women who had been incarcerated in the United States within the past five years (*n* = 20), aged 22–53 years	Semi-structured individual interviews. Grounded theory.	Participants described an institutional culture in which their feminine identity was not recognized and institutional policies acted as a form of structural stigma that created and reinforced the gender binary and restricted access to healthcare. Some participants saw healthcare barriers as a result of bias, others attributed barriers to providers’ limited knowledge or inexperience caring for transgender patients. Access to physical, mental and gender transition-related healthcare negatively impacted participants’ health while incarcerated.	Findings highlight the need for interventions that target multi-level barriers to care in order to improve incarcerated transgender women’s access to quality, gender affirmative healthcare.
Johnson et al. (2020) USA [31]	To describe the spectrum of specific parental behaviours across three categories—rejecting, supportive, and mixed behaviours—and to describe the perceived psychosocial consequences of each of the three categories of parental behaviours on the lives of trans adolescents.	Transgender individuals (*n* = 24), aged 16–20 years.	Two in-depth interviews with each participant. Use of techniques that incorporated visual images and representations. Thematic Analysis.	Overall, participants reported that rejecting and mixed parental behaviours contributed to a range of psychosocial problems (e.g., depression and suicidal ideation), while supportive behaviours increased positive wellbeing.	These findings expand upon descriptions of parental support and rejection within the trans adolescent literature and can help practitioners target specific behaviours for interventions.
Knutson et al. (2018) USA [32]	To explore transgender or gender nonconforming individuals’ health care recommendations for rural settings.	Transgender individuals (*n* = 10), aged 23–59 years.	Semi-structured individual interviews. Consensual Qualitative Research.	Themes looked at access to care, quality control, difficulties, and mentorship. Understanding the content of interpersonal exchanges in transgender communities may support the creation of more effective health services and community building initiatives.	Additional research is needed to assess dimensions of community building and shared knowledge in rural transgender communities that reach beyond healthcare utilization and access.
Lykens et al. (2018) USA [33]	To explore the healthcare experiences of genderqueer or nonbinary young adults.	Genderqueer or non-binary young adults (*n* = 10), aged 20–23 years.	Semi-structured individual interviews. Emergent coding analysis.	Participants faced unique challenges even at clinics specializing in gender-affirming healthcare. They felt misunderstood by providers who approached them from a binary transgender perspective and consequently often did not receive care sensitive to nonbinary identities. Participants felt that the binary transgender narrative pressured them to conform to binary medical narratives throughout healthcare interactions.	GQ/NB young adults have unique healthcare needs but often do not feel understood by their providers. There is a need for existing healthcare systems to serve GQ/NB young adults more effectively.
McCann (2015) Ireland [34]	To elicit the views and opinions of transgender people in relation to mental health concerns.	Transgender individuals (*n* = 4), aged 28–54 years.	Semi-structured individual interviews. Thematic Analysis.	Participants identified challenges and opportunities for enhancing mental health service provision for transgender people and their families. Some of the highlighted concerns related to practitioner attributes and relevant psychosocial supports.	Practitioners need to be knowledgeable and competent in the assessment, diagnosis and treatment of transgender mental health issues. There needs to be adequate funding for future research and collaborative work between transgender community groups and mental health services.
Mizock & Lundquist (2016) USA [35]	To identify the specific issues in the psychotherapy process for transgender or gender non-conforming individuals (TGNC)	Participants who self-identified as TGNC (*n* = 45), aged 21–71 years.	Semi-structured individual interviews. Constant comparative method.	Psychotherapy missteps were identified as education burdening, gender inflation, gender narrowing, gender avoidance, gender generalizing, gender repairing, gender pathologizing, and gate-keeping. Reliance on the client to educate the psychotherapist about trans issues and concerns.	Further research is recommended to focus on the qualities of successful experiences in psychotherapy among TGNC clients to balance this perspective.
Real-Quintanar et al. (2020) Mexico [43]	To explore the medical and mental health needs of transgender men.	Transgender men (*n* = 4), aged 22–39 years.	Semi-structured individual interviews. Thematic analysis.	Participants developed their trans identity in childhood. Many bullied at school. No helpful contact with health professionals reported. Those that did experienced ‘mistreatment’, being criticised and judged. Participants felt shame and rejection.	Health professionals need training and education of trans issues and concerns. They need knowledge, skills and competence to better meet the distinct needs of trans people. Need to address mental health issues caused by stigma and discrimination.
Ritterbusch et al. (2018) Columbia [36]	To examine stigma-related healthcare access and violence experienced by trans women in Columbia.	Transgender women (*n* = 28), aged 19–56 years.	In-depth individual interviews. Constant comparative method.	Some participants (*n* = 7) had experienced life-threatening consequences associated with peer-led injection of liquid silicone or other liquids used for body transformation. Many participants (*n* = 23) reported the informal use of hormone therapy without medical guidance. Some (*n* = 9) started this practice during early adolescence and 20 people reported ‘losing a peer’ to informal body transformations. Many participants (*n* = 20) had experienced torture or other grave human rights violations within the Columbian Healthcare System.	HIV prevention programmes should adopt a human rights process rather than ‘disease control.’ Advocacy and trans support programmes should be rights-based, accessible and safe. Need to be clear policies protecting the rights of trans people and that challenges stigma, victimisation and discrimination. Health curricula should be reflective of the distinct needs of this population and practitioners develop the necessary knowledge, skills and attitudes to provide appropriate supports and interventions.
Sansfaçon et al. (2018) Canada [37]	To explore the factors that influence transgender youths’ wellbeing in Quebec.	Transgender youth (*n* = 24), aged 15–25 years.	In-depth individual interviews. Constant comparative method.	Youth with access to specialized, trans-affirmative health centres and mental health professionals reported how welcoming services and providers helped them affirm their identity and feel supported. Accepting and knowledgeable providers are key to helping youth cope with gender identity issues. A supportive and respectful healthcare environment can contribute to their well-being and health. Supportive psychological services can enhance youth’s capacity to face adversity and build resilience.	Health professionals need to be more knowledgeable about trans issues with clear policies supporting interventions to address possible ‘informational’ and ‘institutional’ erasure. Ensure that any information or knowledge is non-pathologising and non-exclusionary of diversity. Need to apply an intersectional lens to trans experiences that goes beyond gender identity issues alone. Question the distinction between recognition and visibility and what these concepts mean for trans people.
Santos et al. (2019) USA [38]	To elicit transgender university students’ experiences of accessing primary and mental health services via university health services.	Transgender students (*n* = 11), aged 18–24 years.	Semi-structured individual interviews. Thematic analysis.	University Health Services (UHS) are not adequately meeting transgender students’ health care needs. Students reported being repeatedly misgendered and addressed by the incorrect name by staff. Some providers asked inappropriate and irrelevant questions about their gender identity during clinical appointments. These and related experiences deterred many participants from returning to the UHS.	Accurate and inclusive transgender students’ identities are systematically included in their medical records as recommended by the Fenway Institute and WPATH. University Health Services staff should be trained in trans-gender-inclusive best practices and trained in trans-specific health care delivery in order to ensure inclusivity. Provide staff skills training focusing on learning and practicing ways to actively demon-strate both trans-awareness and trans-allyship.
Strauss et al. (2019) Australia [39]	To explore trans and gender diverse young people’s attitudes towards game-based digital mental health interventions.	Trans and gender diverse youth (*n* = 14), aged 11–18 years.	Focus group interviews. Thematic analysis.	Games can bolster general well-being. Peer support can improve mental well-being. Apps were reportedly valuable due to the mental health management skills that they teach the individual, such as coping mechanisms (e.g., through mindfulness, grounding and breathing exercises) and promotion of self-care. Trans informative content is important in game-play. Some containing violence or inappropriate content perceived as unhelpful to mental health.	Game-based digital mental health interventions, and their potential utility in TGD populations have utlility. The intervention should involve TGD or LGBT+ consultation in its development and should be marketed to TGD young people through trusted sources, namely mental health professionals or peers. Trans-affirmative, peer-informed and gender inclusive related content is important to game-play. Participants voiced that a positive feature of many games is the ability to play as, and express, their affirmed gender.
Taylor (2013) Canada [44]	To examine transmen’s experiences of health and social care.	Transmen (*n* = 3), aged 21–29 years.	Semi-structured individual interviews. Thematic analysis.	Major themes indicated provider competence as being problematic in the areas of knowledge gathering, quality helping relationships, and access to health interventions. Cultural competency deficient and a lack of research on trans issues to inform practice. Access to supports and interventions problematic. Self-advocacy common among trans men.	Social workers and other health care providers to expand their thinking beyond binary concepts and move to a more “constellational” view of gender identity. Practitioners have an ethical responsibility to address discrimination based on sex or gender identity, challenge diagnoses such as Gender Identity Disorder and break down the gender binary.
Vermeir et al. (2018) Canada [40]	To identify the barriers to emergency healthcare for trans adults.	Transgender adults (*n* = 8), aged 18–44 years.	Semi-structured individual interviews. Constant comparative method.	Trans participants felt discriminated against and socially excluded in primary and emergency care settings. Discrimination ranged from subtle to overt and often have detrimental consequences. Trans people expected to be more active in their care including educating health professionals.	Important to educate health providers about trans identity to enable trans people to feel more included in their care. Trans narratives should be used to inform future developments in health and social care.
von Vogelsang et al. (2016) Sweden [41]	To describe transgender people’s experiences of health professionals during sexual reassignment process.	Transgender women (*n* = 6), aged 20–36 years.	Semi-structured individual interviews. Content analysis.	Encounters with professionals seen as good when being respectful, acted professionally and built trust and confidence. Poor experiences included lack of knowledge, withholding information, abusing power, gender stereotyping and using the wrong name. Felt dependent on health professionals.	Improved education for health professionals on transgender issues. Increased awareness of the impact of negative attitudes, poor skills and lack of knowledge.
Yan et al. (2019) China [42]	To explore transgender women’s experiences of identity, stigma and HIV in China.	Transgender women (*n* = 14), aged 20–55 years.	In-depth individual interviews and focus group interviews. Thematic analysis.	Participants faced discrimination, poor access to services, unmet mental health needs. Social networks remain sparse and hidden. Almost all participants experienced family rejection. Low awareness and testing for HIV.	Need trans-specific services including gender-affirmative medical and mental health care. HIV prevention strategies required.

## Data Availability

No new data were created or analyzed in this study. Data sharing is not applicable to this article.

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
