# Peer review of "Experiences and Perceptions of Trans and Gender Non-Binary People Regarding Their Psychosocial Support Needs: A Systematic Review of the Qualitative Research Evidence"

_ijerph, 2021, doi:10.3390/ijerph18073403_

Round 1
Reviewer 1 Report
The work presented is of enormous interest and relevance. Even more so at a time when, in many countries, the rights of transgender people are being intensively debated.
On the other hand, we congratulate the authors for using a qualitative perspective in their work.
The work presented is also very careful in its formal and methodological aspects.
Below we present a series of recommendations with the aim of improving the submitted manuscript:
- It would be interesting to expand on how the authors have conducted the thematic analysis. It is not clear in the article.
Author Response
Thank you for your very helpful comments.
Response to your query:
- It would be interesting to expand on how the authors have conducted the thematic analysis. It is not clear in the article.......
Page 3: Data synthesis section has been re-written and expanded upon.
Reviewer 2 Report
The authors have rightly identified that there is scant data describing the needs of and psychological support for TGDNB people. The systematic review of qualitative research is well-written, with few minor grammatical or typographical errors.
The PRISMA-S methodology was appropriately followed. Could the reason for limiting papers to 2010 onwards be clarified?
Although the authors discuss potential ‘discriminatory behaviours and negative attitudes’ from health professionals, it is also important to explicitly recognise and comment on the discredited practice of ‘SOGI change efforts’ published in the literature.
Author Response
Thank you for your helpful comments. Responding to your queries:
- Could the reason for limiting papers to 2010 onwards be clarified?
Page 2 The reasons for limiting papers have been clarified
- ....it is also important to explicitly recognise and comment on the discredited practice of ‘SOGI change efforts’ published in the literature.
Page 14 Reference to 'SOGI' has been included.
Reviewer 3 Report
The objective of this research is to do a systematic review to critically evaluate and synthesize the existing research evidence relating to the socio-psychological experiences of trans and non binary people and identify aspects that may help or hinder access to appropriate psychosocial interventions and supports.
The article is interesting and necessary, it is well written and the methodology is adequate, however it presents some limitations that should be modified before being accepted for publication:
- The authors defend the use of the term trans in the text. In this line, they should also use the term trans (which, on the other hand, is the most appropriate) also in the title and in the abstract.
- The introduction should be rewritten to include non-binary people. In this version it only focuses on trans people.
- Trans people and non-binary people should be included in the objectives
- In Table 1 the search code S3 is missing
- Please check for minor mistakes in the text such as blank spaces (including abstract)
Author Response
Thank you for the helpful comments. Our responses to your queries:
- The authors defend the use of the term trans in the text. In this line, they should also use the term trans (which, on the other hand, is the most appropriate) also in the title and in the abstract.
Checked and amended throughout the manuscript.
- The introduction should be rewritten to include non-binary people. In this version it only focuses on trans people.
Checked and amended through the introduction.
- Trans people and non-binary people should be included in the objectives.
Page 2 Added into the objectives.
- In Table 1 the search code S3 is missing
Table 1: Code added.
- Please check for minor mistakes in the text such as blank spaces (including abstract).
Manuscript checked throughout.
Round 2
Reviewer 3 Report
The authors have made the requested modifications. The manuscript has been significantly improved and now warrants publication in IJERPH.